# A probabilistic description of bedload fluxes from the aggregate dynamics of individual grains

**J. Kevin Pierce**[1], **Marwan A. Hassan**[1], **and Rui M.L. Ferreira**[2]

[1]The University of British Columbia, Vancouver, Canada
[2]Universidade de Lisboa Instituto Superior Técnico, Lisbon, Portugal

**Correspondence:** K. Pierce (j.kevin.pierce@ubc.ca)

**Abstract.** We formulate the bedload sediment flux probability distribution from the Lagrangian dynamics of individual grains. Individual particles obey Langevin equations wherein the stochastic forces driving particle motions are switched on and off by particle entrainment and deposition. The flux is calculated as the rate of many such particles crossing a control surface within a specified observation time. Flux distributions inherit observation-time dependence from the on-off motions of particles. At the longest observation times, distributions converge to sharp peaks around classically-expected values, but at short times, fluctuations are erratic. We relate this scale dependence of bedload transport rates to the movement characteristics of individual sediment grains. This work provides a statistical mechanics description for the fluctuations and observation-scale dependence of sediment transport rates.

## 1 Introduction

Bedload transport refers to conditions when grains bounce and skid along the riverbed (Church, 2006). Numerous applications require predictions of bedload transport rates: ecological restoration, river engineering, and landscape evolution modeling provide examples. Predicting bedload fluxes is notoriously difficult, in part because they display wide fluctuations (Bunte and Abt, 2005; Recking et al., 2012). Variability in transport rates is strongest in weak transport conditions, where instantaneous fluxes can deviate from mean values by several orders of magnitude (Ancey et al., 2008; Benavides et al., 2022). Transport fluctuations lend scale-dependence to measured transport signals, whereby the statistical moments of transport depend on the averaging timescales (Singh et al., 2009; Saletti et al., 2015; Ancey, 2020). Because weak bedload transport conditions are common in gravel-bed rivers (Andrews, 1994; Oldmeadow and Church, 2006; Church and Hassan), predictions of bedload transport should include estimates of variability and the temporal scales over which averaged values converge.

Particle-based, stochastic approaches have been developed from which mean values, probability distributions, and the dependence of measured values on averaging scales can all be obtained (e.g. Ancey and Pascal, 2020; Turowski, 2010).

These approaches provide a more complete description of bedload transport than deterministic models (e.g. Kalinske, 1947; Bagnold, 1966). However, existing stochastic models remain largely kinematic, because they do not model the Newtonian forces generating particle transport (Furbish et al., 2021). In this paper, we develop a new statistical mechanics formulation of the bedload flux which is based on a Newtonian model for the dynamics of individual grains. This effort provides new understanding of how transport rate fluctuations and scale dependence originate from the grain-scale dynamics.

The original stochastic description of bedload displacement is due to Einstein, who calculated particle trajectories as random sequence of rests interrupted by instantaneous steps (Einstein, 1937), in what amounts to a pioneering application of the continuous time random walk (e.g. Montroll, 1964; Weiss, 1994). Many subsequent works have since generalized Einstein's original model, either by introducing different distributions for resting times and step distances than Einstein originally considered (Hubbell and Sayre, 1964; Schumer et al., 2009; Zhang et al., 2012), or by incorporating resting of particles within the bed subsurface (Wu et al., 2020; Pierce and Hassan, 2020b, a). A conceptually different modification of Einstein's model replaces his instantaneous steps with intervals of particle motion at a constant velocity

(Lisle et al., 1998; Lajeunesse et al., 2017). This approach can be summarized with the stochastic equation

$$\dot{x}(t) = V\sigma(t), \tag{1}$$

where $x(t)$ is the sediment position, $V$ is the constant velocity of moving grains, and $\sigma(t)$ is a dichotomous noise which flips randomly between $\sigma = 0$ (rest) and $\sigma = 1$ (motion) (e.g. Bena, 2006). Owing to the constant velocity assumption, this model for sediment displacement applies only at long timescales when the details of individual particle movements become irrelevant (cf. Weiss, 1994). Despite this timescale limitation, Eq. (1) establishes a starting point to describe bedload displacements with additional resolution compared to the random walk approach. To calculate displacements at short timescales, we should replace $V$ with a time-dependent velocity derived from the forces generating particle motions.

The velocities of moving grains fluctuate due to hydrodynamic forcing and particle-bed collisions (Heyman et al., 2016; Fathel et al., 2015). Experiments indicate that downstream velocity distributions of particles can be exponential (Fathel et al., 2015; Lajeunesse et al., 2010), Gaussian (Heyman et al., 2016; Ancey and Heyman, 2014), or Gamma-like (Liu et al., 2019; Houssais et al., 2015). This range of observations has been attributed to differences in the balance of long and short movements within the particle velocity statistics (Wu et al., 2020, 2021), or to differences between experiments in the amount of momentum dissipated by particle-bed collisions (Pierce, 2021). Several studies have successfully modeled the downstream velocity fluctuations of moving particles by making an analogy to Brownian motion (Fan et al., 2014; Ancey and Heyman, 2014). These studies represent exponential and Gaussian velocities with the Langevin equation

$$\dot{v}(t) = F(v) + \xi(t), \tag{2}$$

where $v(t)$ is the downstream sediment velocity, $F(v)$ is a deterministic force per unit mass, and $\xi(t)$ is a random force per unit mass. $\xi(t)$ has been modeled as a Gaussian white noise for simplicity, although this is not an essential requirement. The choice $F(v) = -\mu v/|v| + \Delta$ produces exponential velocities, while $F(v) = \gamma(V - v)$ produces Gaussian velocities. In these expressions, $\mu$ is a Coulomb friction-like coefficient, $\Delta$ is a constant force per unit mass, $\gamma$ is a relaxation rate per unit mass, and $V$ a steady-state velocity. Eq. (2) provides a reasonable description of bedload velocities over short timescales, but it cannot describe particle displacements over longer timescales when entrainment and deposition also moderate the particle transport.

Motion-rest alternation and the fluctuating movement velocities of individual particles both lend variability to the sediment flux (Böhm et al., 2004; Roseberry et al., 2012). Sediment fluxes have been defined with both surface and volume definitions (Ballio et al., 2014, 2018). The *ensemble averaged* flux has been formulated using a surface definition by

Furbish et al. (2012, 2017):

$$\langle q(x,t) \rangle = \int\limits_{0}^{\infty} dx' \int\limits_{0}^{\infty} dt' R(x',t') E(x-x',t-t'). \tag{3}$$

This "nonlocal formulation" generalizes earlier approaches based on aggregating particles from upstream locations (Nakagawa and Tsujimoto, 1976; Parker et al., 2000). In this equation, $E(x-x',t-t')$ is the average entrainment rate a distance $x'$ upstream and a time $t'$ in the past, while $R(x',t')$ is the probability that a just-entrained particle displaces *at least* a distance $x'$ in time $t'$. For the steady case $E(x,t) = E$ where particles step an average distance $\ell$ between entrainment and deposition, the nonlocal flux becomes $\langle q \rangle = E\ell$, in accord with Einstein (1950). The nonlocal formulation has not yet been extended to characterize the sediment flux as a stochastic process.

The sediment flux has been described as a stochastic process using both renewal theory and population modeling approaches. These methods rely on additional Eulerian characteristics of particle transport which apply to an ensemble of particles occupying a volume or crossing a surface. Renewal models introduce inter-arrival time distributions $\psi(t)$ which characterize the time intervals between successive arrivals of particles to a surface (Turowski, 2010; Ancey and Pascal, 2020). The flux is calculated as the rate of particle crossings over an observation time $T$:

$$q(T) = \frac{\mathcal{N}(T)}{T}, . \tag{4}$$

Here, $\mathcal{N}(T)$ is the number of particle crossings by $T$ – related to $\psi(t)$. For exponential inter-arrival times $\psi(t) = \lambda e^{-\lambda t}$, the flux becomes Poissonian with *rate constant* $\lambda$, so the mean flux is $\langle q \rangle = \lambda$. For other $\psi(t)$'s, the flux depends on the observation time $T$ in a type of scale dependence. Instead of counting particle arrivals, population modeling approaches introduce Eulerian entrainment and deposition rates to count the number of moving particles in a volume (Ancey et al., 2006, 2008). These approaches compute the flux by summing the velocities of all moving particles in a volume (Heyman, 2014; Ancey and Pascal, 2020). Both of these stochastic approaches provide excellent correspondence with experimental data. However, we still have little understanding of how to relate their Eulerian input parameters to the underlying grain-scale mechanics (Heyman et al., 2016). This limitation exposes a need for further research into how the trajectories of individual grains ultimately produce a stochastic sediment flux and control its dependence on observation scale.

In this paper, our objective is to formulate the stochastic sediment flux directly from the Lagrangian dynamics of individual grains, rather than by introducing additional Eulerian quantities, such as volumetric entrainment and deposition rates or inter-arrival time distributions. To achieve this,

in Sec. 2 we extend the motion-rest alternation model for particle displacement, Eq. (1) to include Newtonian velocities from Eq. (2). This produces a mechanistic-stochastic model of particle displacement which is valid across a wider range of timescales than earlier descriptions. We then construct the sediment flux in Sec. 3 by accumulating the displacements of individual particles. The resulting formulation shares elements from both the nonlocal and renewal approaches summarized in Eqs. (3) and (4). In Sec. 4 we solve the formulation to derive the probability distributions of particle position and the sediment flux, and in Secs. 5 and 6 we discuss the new features and limitations of our approach, summarize its relationship to earlier work, and suggest several directions for further development.

## 2 Stochastic description of bedload transport

The starting point for our analysis is an idealized one-dimensional domain populated with sediment particles on the surface of a sedimentary bed. Particles are set in motion by the flow and move downstream until they deposit, and the cycle repeats. The downstream coordinate is $x$, so that $\dot{x} = dx/dt$ describes a velocity in the downstream direction. The flow is considered weak enough that interactions among moving grains are uncommon, although interactions between moving particles and the bed occur regularly. These conditions are characteristic of "rarefied" bedload transport conditions (e.g. Kumaran, 2006; Furbish et al., 2017). Particles are considered to have similar enough shapes and sizes so as to have nearly identical mobility characteristics. These conditions allow for all particles to be described as independent from one another but governed by the same underlying dynamical equations. Any of these conditions could be relaxed in exchange for additional mathematical difficulty.

### 2.1 Mechanistic formulation of intermittent transport

From these assumptions, we propose an equation of motion for the individual sediment grain including two features. First, particles should alternate between motion and rest, similar to the earlier motion-rest models summarized by Eq. (1). Second, the velocities of moving particles should evolve according to the Newtonian equation (2). These dynamics can be represented as

$$\dot{x}(t) = v(t)\sigma(t),$$
$$\dot{v}(t) = [F(v) + \xi(t)]\sigma(t). \tag{5}$$

In these equations, $F(v)$ is a deterministic forcing term whose structure can be chosen to produce the desired velocity distribution for moving particles (exponential, Gaussian, or others), $\xi(t)$ is a Gaussian white noise with correlation function

$$\langle \xi(t)\xi(t+\tau) \rangle = 2\Gamma\delta(\tau) \tag{6}$$

that represents the velocity fluctuations of moving particles, and $\Gamma$ is a velocity diffusivity [units $L^2/T^3$] which controls the intensity of velocity fluctuations. Fig. (1) displays particle velocities and displacements derived from these equations for the choice of Gaussian movement velocities.

These equations represent a Newtonian dynamics that is *randomly paused* as $\sigma(t)$ alternates. The quantity $\sigma(t)$ is a dichotomous Markov noise (e.g. Horsthemke and Lefever, 1984; Bena, 2006) which produces alternation between motion ($\dot{x}$ generally nonzero) and rest ($\dot{x}$ zero). This noise takes on values $\sigma = 1$ (motion) and $\sigma = 0$ (rest). The entrainment rate ($\sigma = 0 \rightarrow \sigma = 1$) is labeled $k_E$, and the deposition rate ($\sigma = 1 \rightarrow \sigma = 0$) is labeled $k_D$. Times spent in motion and rest are respectively distributed as $P(t) = k_D \exp(-k_D t)$ and $P(t) = k_E \exp(-k_E t)$, so the mean movement time is $k_D^{-1}$, and the mean resting time is $k_E^{-1}$. The notation $k = k_E + k_D$ is a shorthand used throughout the paper. $1/k$ represents the correlation time of the dichotomous noise.

The transport process described by Eq. (5) is intermittent because the particle velocity $\dot{x}$ and acceleration $\dot{v}$ are randomly switched on and off by entrainment and deposition. Notably, the presence of $\sigma(t)$ in these two equations *decouples* $\dot{x}$ and $v$. In general, $\dot{x} \neq v$, in contrast to conventional mechanics. This decoupling means the relevant velocity scale for particle displacement is the "virtual velocity" $\dot{x}$. This velocity is "virtual" because it contains the particle velocity during motions in addition to the intervening rests (cf. Hassan et al., 1991; Bradley and Tucker, 2013). The second velocity scale $v$ represents the velocities of particles *if* they are moving. During motions, $v$ evolves as a stochastic process. This evolution is paused during particle rests, when deposition shuts off the driving forces in Eq. (5).

The formulation of Eq. (5) is conceptually similar to the work of Fan et al. (2016), which simulates particle transport by manually switching Eq. (2) on and off to represent entrainment and deposition. We have replaced this manual switching with a dichotomous noise to obtain an analytical representation of particle transport by motion-rest alternation. This description is similar to several intermittent transport models in the stochastic physics literature, as it involves both (1) interrupted diffusion (e.g. Laskin, 1989; Łuczka et al., 1993; Balakrishnan et al., 2001) and (2) a dichotomous noise entering at second order in time (e.g. Masoliver, 1992, 1993). To our knowledge, Eq. (5) is the first stochastic transport to include both of these components simultaneously.

### 2.2 Phase space description of motion-rest alternation

The time evolution of Eq. (5) for a particular realization of $\xi(t)$ and $\sigma(t)$ maps a trajectory through the phase space spanned by $x$ and $v$. The conditional probability density $W(x, v, t | x_0, v_0)$ represents the likelihood that a phase trajectory reaches $(x, v)$ at time $t$ provided it passed through $(x_0, v_0)$ at $t = 0$. This density characterizes the stochastic evolution of the particle position and velocity.

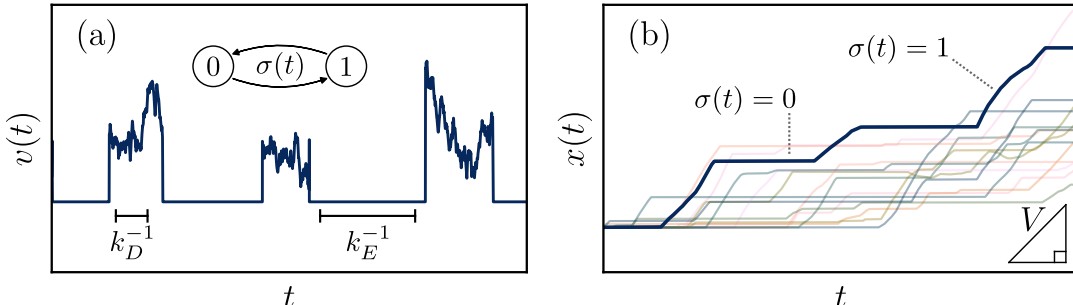

**Figure 1.** Panel (a) shows the velocity from Eq. (5) for a particular realization of the noises $\xi(t)$ and $\sigma(t)$, while panel (b) shows the position derived Eq. (5) alongside other possible trajectories. Keys in panel (a) indicate the average movement time $1/k_D$, rest time $1/k_E$, and motion-rest alternation process of $\sigma(t)$, while keys in panel (b) shows the average movement velocity $V$ and evolution of the displacement for different values of $\sigma(t)$. Motion-rest alternation with velocity fluctuations produces tilted stair-step trajectories with unsteady slopes in the $x$–$t$ plane.

A master equation for the phase space density can be formulated by noting that the combined process $(x, v, \sigma)$ is Markovian (cf. Horsthemke and Lefever, 1984). In appendix A, we demonstrate that Eq. (5) implies the master equation

$$\partial_t(\partial_t + k)W(x, v, t|0) = (\partial_t + k_E)\hat{L}_K W(x, v, t|0), \quad (7)$$

using the abbreviation $W(x, v, t|x_0, v_0) = W(x, v, t|0)$. In this equation, $\hat{L}_K$ is the *Kramers operator*, familiar from the description of Brownian particles in an external force field (e.g. Risken, 1984; Kubo et al., 1978):

$$\hat{L}_K = -v\partial_x + \partial_v\{-F(v) + \Gamma\partial_v\}. \quad (8)$$

The Kramers operator evolves the particle position and velocity. The term with $\partial_x$ represents drift, while terms containing $\partial_v$ represent accelerations, with advective contributions from $F(v)$ and diffusive ones from $\Gamma$ (Duderstadt and Martin, 1979).

To understand the structure of Eq. (7), it is useful to recall the classical Kramers equation for particles driven by a fluctuating force without intermittency: $\partial_t W(x, v, t|0) = \hat{L}_K W(x, v, t|0)$, with the same operator $\hat{L}_K$ as above (Kramers, 1940). A comparison suggests that the additional terms in Eq. (7) weave intermittency into the distribution function. In fact, there are two limiting behaviors contained in Eq. (7) at opposite extremes of $t$ (cf. Bena, 2006). At short times, higher orders of $\partial_t$ dominate, giving the classical Kramers equation $\partial_t W \approx \hat{L}_K W$ after an integration over time. At long times, the lower order terms in $\partial_t$ dominate instead, giving $\partial_t W \approx (k_E/k)\hat{L}_K W$. This is still a Kramers-type equation, although the evolution of $x$ and $v$ are *slowed* by the "intermittency factor", $k_E/k$. This factor represents the average fraction of time particles have spent in motion. Thus the higher order time derivatives in Eq. (7) encode a transition between conventional and intermittent regimes of

motion. This structure reflects the increasing influence of entrainment and deposition on the evolution of $x$ and $v$ with time.

### 2.3 The displacement statistics of bedload grains

To calculate the sediment transport rate we will use the probability distribution of position for bedload particles, defined as

$$P(x, t|0) = \int_{-\infty}^{\infty} dv' W(x, v', t|0). \quad (9)$$

Unfortunately, even for the classical Kramers equation without intermittent motion, it is extremely difficult to obtain a governing equation for $P(x, t|0)$ without first solving the phase space master equation for $W(x, v, t|0)$ (e.g. Brinkman, 1956; Olivares-Robles and García-Colín, 1996). For the case of $F(v)$ associated with exponential velocities (e.g. Fan et al., 2014), the Kramers equation has only been solved numerically (Menzel and Goldenfeld, 2011).

Fortunately, bedload experiments often show Gaussian velocities for moving particles (e.g. Martin et al., 2012; Ancey and Heyman, 2014; Heyman et al., 2016), corresponding to the forcing term

$$F(v) = \gamma(V - v) \quad (10)$$

in Eq. (7). With this force, the classical (non-intermittent) Kramers equation analogous to Eq. (7) can be solved exactly for the position distribution $P(x, t|0)$ (e.g. Wang and Uhlenbeck, 1945; Chandrasekhar, 1943).

When motions are intermittent as in Eq. (7), an analytical solution appears not yet possible. However, we can still solve Eq. (7) with the force (10) approximately by using the same "overdamped" approximation often applied to the classical Kramers equation (e.g. Risken, 1984; Gardiner, 1983).

This approximation is possible whenever a term like Eq. (10), linear in velocity, is present in a governing Langevin equation. In our context, the overdamped approximation applies whenever moving particles attain their steady-state velocities relatively quickly after entrainment. Campagnol et al. (2015) suggests that the "acceleration phase" following entrainment will only affect a small portion of the trajectory between motion and deposition when the flow is sufficiently strong. In addition, earlier Langevin models have successfully described the velocity distributions of bed load particles by neglecting the transient acceleration phase following entrainment (Fan et al., 2014; Ancey and Heyman, 2014). For these reasons, we expect that the overdamped approximation is a reasonable first approach for solving Eq. (7).

We can actually obtain the overdamped approximation for the phase space equation (7) using the same method originally introduced by Kramers (1940). We integrate Eq. (7) along the straight line $x + v\gamma^{-1} = \text{const.}$ from $v \to -\infty$ to $v \to \infty$. Because $\gamma^{-1}$ is small for fast relaxation, we can take the line integral along $dv$ only (cf. Coffey et al., 2004), providing the overdamped master equation:

$$\left[\partial_t^2 + k\partial_t + V\partial_x\partial_t + k_E V\partial_x - D\partial_x^2\partial_t - k_E D\partial_x^2\right]P(x,t) = 0. \tag{11}$$

The spatial diffusivity $D$ is defined as $D = \gamma^{-2}\Gamma$ with units $[L^2/T]$. Hereafter we suppress the explicit dependence of the position distribution on its initial conditions $[P(x,t|x_0,v_0,t_0) = P(x,t)]$.

Eq. (11) interleaves two different diffusion processes: one associated with motion-rest alternation, and another with velocity fluctuations during motions. Terms involving $V$ represent advection, while those involving $D$ represent diffusion from velocity fluctuations. Terms involving $k_E$ and $k$ represent diffusion due to motion-rest alternation. Mixed orders of $\partial_t$ encode the aforementioned transition between conventional and intermittent transport.

## 3   Mechanistic formulation of the sediment flux

To phrase the probability distribution of the sediment flux in terms of these particle dynamics, we apply a method very similar to the one developed by Banerjee et al. (2020) to describe currents of active particles in condensed matter physics. The basic idea, as depicted in Fig. (2), is to initially distribute $N$ particles in all states of motion along the domain $-L \leq x \leq 0$. Later, the number of particles $N$ and the size $L$ of the domain will be extended to infinity such that their ratio $\rho = N/L$ remains constant. This limit produces a configuration similar to the one considered in the nonlocal formation of Eq. (3).

From this initial configuration, the flux is calculated as the average rate of particles crossing to the right of the control surface at $x = 0$ after the sampling time $T$, analogous to the renewal formulation of Eq. (4):

$$q(T) = \frac{1}{T}\sum_{i=1}^{N} I_i(T). \tag{12}$$

In this equation, the $I_i(T)$ are indicator functions which equal 1 if the $i$th particle has crossed the control surface ($x = 0$) by $T$, and 0 otherwise. Particles which have not crossed the surface (or which have crossed and then crossed back) do not contribute to the flux.

The probability density $F(q|T)$ of the flux, conditional on the sampling time $T$, is then an average involving Eq. (12) across all possible initial configurations of particles and their trajectories:

$$F(q|T) = \left\langle \delta\left(q - \frac{1}{T}\sum_{i=1}^{N} I_i(T)\right)\right\rangle. \tag{13}$$

Taking the Laplace transform $\tilde{F}(s|T) = \int_0^\infty dq e^{-sq}F(q|T)$ (forming the characteristic function) produces

$$\tilde{F}(s|T) = \prod_{i=1}^{N}\left[1 - \left(1 - e^{-s/T}\right)\langle I_i(T)\rangle\right]. \tag{14}$$

This formula relies on the independence of averages for each particle (so that the average of a product is the product of averages) and the observation that $e^{\alpha I} = 1 - (1 - e^\alpha)I$ if $I = 0, 1$.

The average over initial conditions and possible trajectories of the indicator for the $i$th particle involved in this characteristic function is

$$\langle I_i(T)\rangle = \frac{1}{L}\int\limits_{-L}^{0} dx' \int\limits_{0}^{\infty} dx P(x - x', T). \tag{15}$$

Here $P(x,T)$ is the position probability distribution of position at time $T$, either derived exactly from Eqs. (7) and (9), or from the overdamped approximation Eq. (11). The integral over $x$ evaluates the probability that the position of a particle at $T$ exceeds $x = 0$, while the integral over $x'$ averages across a uniform distribution of possible initial positions. These $\langle I_i(t)\rangle$ are the components of the flux that depend on the particle dynamics.

Inserting Eq. (15) into Eq. (14) and taking the limits $L \to \infty$ and $N \to \infty$, as the density of particles $\rho = N/L$ is held constant, provides

$$\tilde{F}(s|T) = \exp\left[-\left(1 - e^{-s/T}\right)\Lambda(T)\right]. \tag{16}$$

$\Lambda(T)$ is the central parameter of the sediment flux probability distribution:

$$\Lambda(T) = \rho \int\limits_{0}^{\infty} dx \int\limits_{0}^{\infty} dx' P(x + x', T). \tag{17}$$

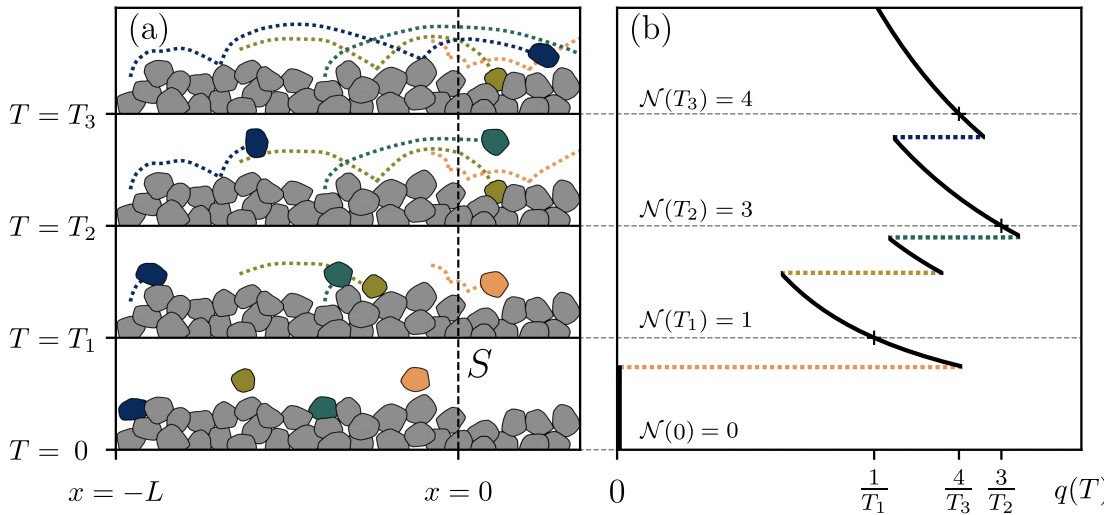

**Figure 2.** Panel (a) indicates the configuration for the flux. Here, time increases from the bottom to the top. Particles begin their transport with positions $-L \leq x \leq 0$ to the left of $S$ at observation time $T = 0$, and the flux is calculated in panel (b) as the rate $\mathcal{N}(T)/T$ of particles crossing $x = 0$ over the observation time $T$. Particle crossing events are indicated in (b) by color-coded lines. The probability distribution of $q(T)$ is determined from all possible realizations of the trajectories and initial positions as $N$ and $L$ tend to infinity while the density of particles $\rho = N/L$ to the left of $S$ is held constant.

The quantity $\Lambda(T)/T$ is a *rate function*. This ratio describes the rate of particle arrivals to the control surface at $x = 0$, and it explicitly depends on the observation time $T$.

Eq. (16) is the characteristic function of a Poisson distribution (Cox and Miller, 1965). Expanding in $e^{-s/T}$ and inverting the Laplace transforms provides the probability distribution of the flux conditional on the sampling time $T$:

$$F(q|T) = \sum_{n=0}^{\infty} \frac{\Lambda(T)^n}{n!} e^{-\Lambda(T)} \delta\left(q - \frac{n}{T}\right). \tag{18}$$

This equation implies that the mean flux is $\langle q(T) \rangle = \int_0^{\infty} qF(q|T)dq = \Lambda(T)/T$. Similarly the variance is $\sigma_q^2(T) = \Lambda(T)/T^2$. For the case when $\Lambda(T)$ is proportional to the observation time ($\Lambda \propto T$), these formulas become identical to the renewal approach with exponential inter-arrival times.

Eq. (18) formulates the flux probability distribution directly from the particle dynamics set out in Eq. (5). This equation is a scale-dependent Poisson distribution. The Poisson form originates primarily from the assumption that particles undergo independent dynamics. The distribution is scale dependent through the displacement statistics of individual particles ingrained in Eq. (17).

## 4 Results

### 4.1 Displacement by intermittent transport

The overdamped master equation (11) describes the displacement statistics of particles alternating between motion and rest with Gaussian movement velocities. This equation is founded on the approximation that just-entrained particles attain their steady-state (but fluctuating) velocities rapidly.

The overdamped master equation (11) is solved in appendix B with transform calculus, obtaining

$$P(x,t) = \hat{A} \int_0^t dt' \, e^{-k_E(t-t')-k_D t'}$$

$$\times \mathcal{I}_0\left(2\sqrt{k_E k_D t'(t-t')}\right) G(x - Vt', t'). \tag{19}$$

Here $\mathcal{I}_0$ is a modified Bessel function, $\hat{A} = -D\partial_x^2 + V\varphi\partial_x + k + \partial_t$ is a differential operator, and $G(x,t) = \exp\left[-x^2/4Dt\right]/\sqrt{4\pi Dt}$ is a Gaussian propagator. Within the integral, the Bessel term represents the proportion of time $u/t$ the particle has spent in motion, while the Gaussian term describes the distribution of displacements achieved in time $u$. This distribution is compared with numerical simulations of the exact distribution from Eq. (5) in Fig. 3a. The general decreasing trend of mean transport with observation time is qualitatively consistent with laboratory observations (Singh et al., 2009; Saletti et al., 2015) and the renewal approach summarized earlier (Turowski, 2010; Ancey and Pascal, 2020).

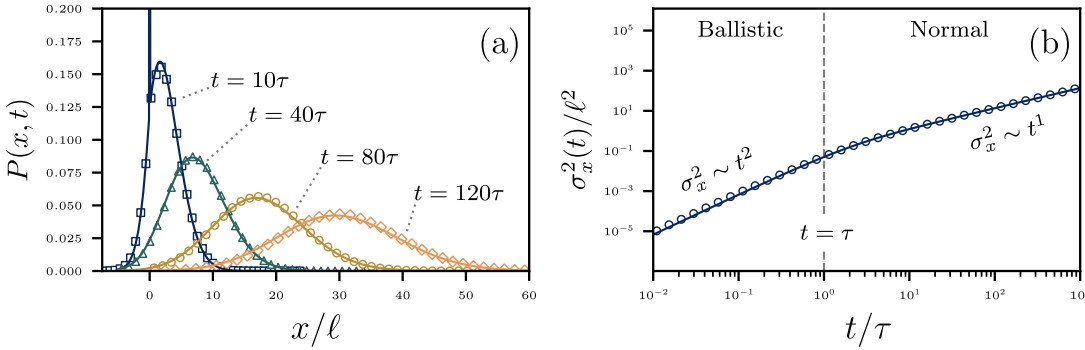

**Figure 3.** Panel (a) indicates the overdamped probability distribution of particle position (19) as it evolves through time, while panel (b) shows the resulting particle diffusion from Eq. (20). All results are scaled by the mean hop length $\ell = V/k_D$ and the timescale $\tau = 1/k$ of the motion-rest alternation. Curves represent the analytical results while the points are results of Monte Carlo simulations of the exact equations (5) produced by evaluating the cumulative transition probabilities on a small timestep (e.g. Barik et al., 2006). In panel (a) from the initial mixture of motion and rest states, particles advect downstream as they diffuse apart from one another due to motion-rest alternation and velocity variations. The initial distribution persists as a delta function spike for the $t = 10\tau$ curve. In panel (b) at timescales $t \ll 1/k$ the diffusion is approximately ballistic since particles have not had time to exchange between motion and rest. For $t \gg 1/k$ particles undergo normal diffusion as particles become well-mixed among motion and rest states. Small discrepancies are visible at short times between the simulations and analytical approximations due to our neglect of velocity fluctuations in deriving Eq. (20).

The moments of particle position from Eq. (5) are challenging to obtain. An approximation for the mean can be obtained by considering that velocity fluctuations during motions approximately cancel out, since these are symmetrical around $V$. Therefore we set $\Gamma = 0$ in Eq. (5), multiply by $x$, and integrate to find the mean position $\langle x(t) \rangle = k_E V t/k$, which is $Vt$ scaled by the expected fraction of time spent in motion. Similarly we can approximate the variance by reasoning that motion-rest alternation, and not velocity fluctuations during motions, is the primary source of particle diffusion. Again setting $\Gamma = 0$ we find the variance of position ($\sigma_x^2 = \langle x^2 \rangle - \langle x \rangle^2$):

$$\sigma_x(t)^2 = \frac{2k_E k_D V^2}{k^3}\left(t + \frac{1}{k}e^{-kt} - \frac{1}{k}\right). \tag{20}$$

This describes a two-range particle diffusion process, whereby the rate of particle spreading depends on how long the dynamics have been ongoing (Taylor, 1920). Fig. (3)b compares this variance to the numerical simulations. The variance exhibits a crossover between ballistic and normal scaling at $\tau = 1/k$. The approximate variance calculation is good apart from small undershooting at short times. Here, small contributions to the variance from velocity fluctuations in the motion state become visible, consistent with the expectation that particle velocity fluctuations during motions will enhance the particle diffusion at short timescales.

## 4.2  The flux rate function for overdamped transport

The formalism in Sec. 3 provides the central parameter $\Lambda(T)$ of the sediment flux distribution. Using the overdamped

probability distribution of position Eq. (19), we evaluate Eq. (17) in appendix C, providing

$$\Lambda(T) = \rho \int_0^T dT'\, \mathcal{I}_0\Big(2\sqrt{k_E k_D T'(T-T')}\Big)e^{-k_E(T-T')-k_D T'}$$

$$\times \left[ \sqrt{\frac{D}{\pi T'}}\Big([\overleftarrow{\partial}_T + k]T' - \frac{k_D}{2k}\Big)e^{-V^2 T'/4D} \right.$$

$$\left. + \frac{V}{2}\Big([\overleftarrow{\partial}_T + k]T' - \frac{k_D}{k}\Big)\mathrm{erfc}\left(-\sqrt{\frac{V^2 T'}{4D}}\right) \right]. \tag{21}$$

In this equation, the notation $\overleftarrow{\partial}_T$ means that the partial time derivative acts from the left of all terms in which it is involved, as in $f(T)\overleftarrow{\partial}_T g(T) = \partial_T[f(T)g(T)]$, and $\mathrm{erfc}(\cdot)$ denotes the complementary error function.

This result indicates a nuanced observation-scale dependence in the sediment flux. We can better understand Eq. (21) by investigating extreme cases of the observation time. As shown in appendix D, Eq. (21) takes on simple forms at extreme values of $T$:

$$\Lambda(T) = \begin{cases} \frac{\rho k_E}{k}\sqrt{\frac{DT}{\pi}}, & T \ll (k_D Pe)^{-1}, \\ \frac{\rho k_E V T}{k}, & T \gg (k_D Pe)^{-1}. \end{cases} \tag{22}$$

Here, the quantity $Pe = V^2/(2Dk_D)$ is a Péclet number which measures the relative importance of advection and diffusion during particle motions. After entrainment, particles typically move for a duration $1/k_D$ before deposition, over which they will displace $L_A = V/k_D$ by advection and

$L_D = \pm\sqrt{2D/k_D}$ by diffusion. $Pe$ is composed as a squared ratio of these length scales: $Pe = (L_A/L_D)^2$.

The limiting form of $\Lambda(T)$ implies that for $T \gg (k_D Pe)^{-1}$ the mean flux converges to the eventual value $\langle q(T \to \infty) \rangle = q_0$:

$$q_0 = \rho k_E V/k. \tag{23}$$

This can be understood as the result $q_0 = E\ell$ of the nonlocal formulation (3) in the case of steady transport conditions. Thus at $T \to \infty$, our formlation becomes equivalent to that of Einstein (1950). Here, $E = \rho k_E$ is an averaged areal entrainment rate and $\ell = V/k \approx V/k_D$ is the mean step length of particles. $k \approx k_D$ holds since the mean duration of a single motion ($1/k_D$) is typically much smaller than the duration of a single rest ($1/k_E$).

Fig. 4a shows the rate constant decaying toward its asymptotic value in Eq. (22) for different values of $Pe$, with all other parameters fixed. Numerical simulations of the exact equations (5) are superimposed. The overdamped approximation pursued in this paper provides a valid characterization of the sediment flux for $T \gg 1/\gamma$, but for $T \ll 1/\gamma$, the approximate result overshoots. Thus we expect that the particle acceleration phase immediately after entrainment (cf. Campagnol et al., 2015) slows the observation-scale dependence of the flux at short timescales.

Fig. 4b demonstrates the adjustment of the flux distribution (12) across observation times. At the shortest times, the flux approaches a uniform-like distribution due to the (apparent) divergence of $\Lambda(T)$. At very long observation times, the flux adopts the deterministic (Einstein) form

$$F(q|T) \sim \delta(q - E\ell). \tag{24}$$

This limit can be seen by taking large $T$ in Eq. (18) and using the correspondence between Poisson and Gaussian distributions for large Poisson rates (e.g. Cox and Miller, 1965). The Einstein (1950) result for the sediment transport rate becomes exact only when $k_D T \gg Pe^{-1}$. Otherwise, the flux has intrinsic fluctuations related to the unpredictability of particle arrivals in a finite observation window, as characterized by Eq. (12).

## 5 Discussion

In this paper, we have formulated a mechanistic description of the bedload sediment flux using a detailed stochastic model of individual particle displacements. The resulting sediment flux distribution shows Poissonian fluctuations that depend on observation scale. Our displacement model applies over a wider range of timescales than earlier formulations because it includes both Newtonian velocities and motion-rest alternation. In appropriate simplified limits, the displacement model Eq. (7) reduces to many earlier descriptions of grain-scale transport (e.g. Einstein, 1937; Lisle et al., 1998; Lajeunesse et al., 2017; Ancey and Heyman, 2014; Fan

et al., 2014). In this sense, the grain-scale transport component of this paper generalizes and unifies all of these earlier works.

We solved the displacement model analytically to obtain the displacement probability distribution. This derivation relied on the "overdamped" approximation that particles accelerate rapidly following entrainment. This approximation is only possible when the velocities of moving grains are Gaussian. We then formulated the stochastic sediment flux using the resulting particle displacement statistics. The obtained flux distribution mimics earlier renewal theory descriptions of the bedload flux (Turowski, 2010; Ancey and Pascal, 2020), except its input parameters relate directly to the Lagrangian transport characteristics of individual grains. The scale dependence of the flux is mainly controlled by the Péclet number and deposition rate of grains, indicating that the scale dependence originates from the velocity fluctuations and motion-rest alternations of individual grains. The flux is enhanced at short observation times because it is dominated by uniquely fast-moving particles. As time grows, a plateau may emerge if the diffusion is sufficiently strong. This plateau can likely be attributed to slow moving particles "catching up" with grains which were faster to cross the control surface, slowing the flux decay. When the time spent in motion goes to zero (idealized steps), or velocity fluctuations vanish ($Pe \to \infty$), the flux loses its scale dependence. In other conditions, as the observation time becomes large, the flux approaches the deterministic result $q = E\ell$ of Einstein (1950) and later nonlocal formulations (Furbish et al., 2012, 2017).

### 5.1  Newtonian description of bedload displacement

Our description of individual particle displacements in Sec. 2 provides an analytically-solvable alternative to computational-physics models of grain-scale transport (e.g. Schmeeckle, 2014; Ji et al., 2014; Clark et al., 2017). Analytical progress was possible largely because the forces on moving particles were modeled as linear in the particle velocity [Eq. (10)] with Gaussian white noise fluctuations (cf. Ancey and Heyman, 2014). This forcing structure is a crude approximation for the actual hydrodynamic and granular forces acting on bedload particles. In reality, flow forces are non-linear in the flow velocity and history-dependent (Michaelides, 1997; Schmeeckle et al., 2007), while collision forces are velocity-dependent and episodic (Brach, 1989; Schmeeckle and Nelson, 2003; Pierce, 2021). Fluctuations in turbulent flow forces are *colored* noise, not white (Schmeeckle et al., 2007; Dwivedi et al., 2010; Celik et al., 2014). Although we cannot hope to include completely realistic forces into an analytically-solvable model, it should still be possible to introduce some level of additional complexity into the forcing structure of Eq. (5). A first step is to solve Eq. (7) with $F(v)$ tailored to produce exponential velocities for moving particles (e.g. Fan et al., 2014). The result would lend additional

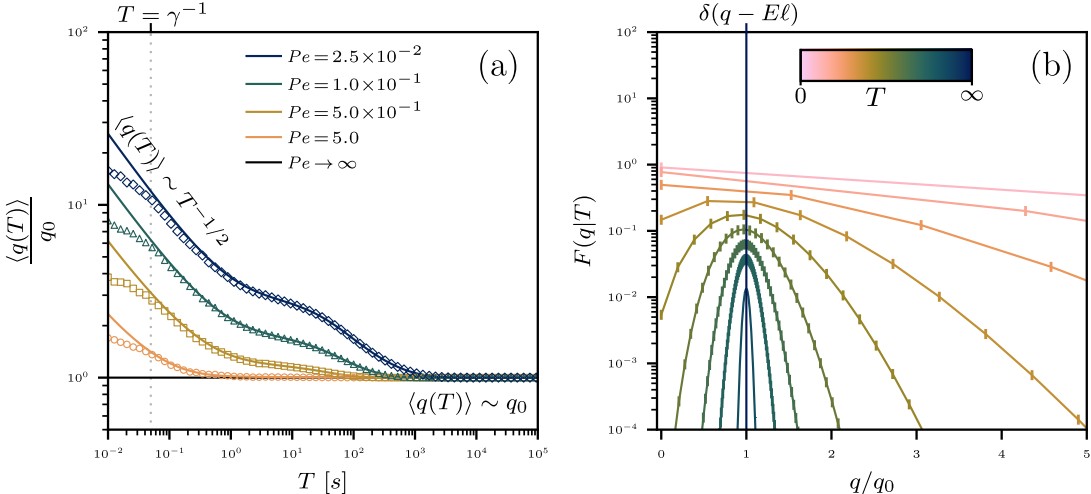

**Figure 4.** Panel (a) plots the mean sediment flux for different values of the Péclet number $Pe = V^2/(2k_D D)$, characterizing the relative significance of particle velocity fluctuations during motions. Plotted lines show the analytical result (21), while the points are Monte Carlo simulations. Panel (b) displays the evolution of the flux distribution (18) across observation times. The flux is normalized by the prediction $q_0 = E\ell$ of the Einstein model. In panel (a), the convergence time of the mean flux is controlled by attributes of individual particle motions. In all cases, the mean flux converges for $T \gg 1/(k_D Pe)$, as expected by Eq. (22). Discrepancies between numerical simulations and analytical calculations emerge for timescales $T \ll \gamma^{-1}$, indicating that the overdamped approximation cannot account for this "acceleration phase" period. A plateau emerges at intermediate times when velocity fluctuations are strong ($Pe \ll 1$). This plateau is likely associated with particles whose displacements lag behind due to diffusion "catching up" to other crossing grains. In panel (b), the Einstein limit $F(q|T) = \delta(q - E\ell)$ is approached as the observation time $T$ grows. Stronger velocity fluctuations (smaller $Pe$) slow the convergence. In all plots, $Pe$ is modified by changing $D$, while all other parameters are held constant.

insight into how the scale-dependence of the flux depends on the particle velocity statistics.

A simplified element of our approach is the representation of entrainment and deposition by instantaneous alterna-
5 tion between motion and rest states (e.g. Einstein, 1937). In actuality, motion and rest are not perfectly defined because "resting" particles undergo slow downstream creep (Houssais et al., 2015; Allen and Kudrolli, 2018) and can shuttle in their pockets without any net translation (Diplas et al., 2008;
Celik et al., 2010). Our dichotomous representation of entrainment and deposition provides no direct information on the timescales of particle motion and rest, and relating the motion and rest timescales of grains to the underlying fluid and granular mechanics remains an important open problem.
It is challenging to imagine an analytically-solvable model of particle displacement which does not discriminate between multiple states of transport, but improvements can nonetheless be made to the dichotomous representation of entrainment and deposition we employed. One option is to render
slower-moving particles more likely to deposit, which can be viewed as making the dichotomous noise "state dependent" (e.g. Laio et al., 2008; Bartlett and Porporato, 2018). This would preferentially cull slow-moving particles and positively skew the particle velocity distribution (Williams and
Furbish, 2021), probably affecting the sediment flux.

## 5.2   Mechanistic interpretation of transport fluctuations

The sediment flux probability distribution derived in Sec. 3 represents a Poisson distribution with a scale-dependent rate. Poisson distributions have relatively thin tails which reflect narrow sediment transport fluctuations. In reality, sediment 30 flux distributions are only Poissonian at high transport rates, whereas in other conditions they have wide tails representing the possibility of large fluctuations (Ancey et al., 2008; Saletti et al., 2015; Turowski, 2010), which appear as bursts (e.g. Goh and Barabási, 2008) in sediment flux timeseries 35 (Dhont and Ancey, 2018; Singh et al., 2009). This observation highlights a need to improve the mechanistic description we developed here to produce wider transport rate fluctuations.

A vast set of processes generates transport rate fluctua- 40 tions in real channels. At the shortest timescales, fluctuations arise from the intermittent arrivals of individual grains, as we have described here. Over longer timescales, activity waves (Heyman et al., 2014), cluster entrainment (Papanicolaou et al., 2018; Strom et al., 2004), bedform migra- 45 tion (Guala et al., 2014; Hamamori, 1962), grain-size sorting (Cudden and Hoey, 2003; Iseya and Ikeda, 1987), flow variations (Mao, 2012; Wong and Parker, 2006), and sediment supply perturbations (Lisle et al., 1993; Madej et al., 2009) all contribute to sediment transport variability. It should be 50 possible to include particle-particle interactions into our de-

scription to capture the subset of these processes which originate at the grain scale. Activity waves, clusters, and bedforms might result from including interactions between particles into the entrainment and deposition rates, such as collisions (Lee and Jerolmack, 2018), the stabilization of bed particles by neighbors (Church et al., 1998), or coordinated deposition based on the locations of sedimentary deposits (McDowell and Hassan, 2020). We might formulate the resulting joint distribution of particle positions and velocities by analogy to reaction-diffusion problems (e.g. Pechenik and Levine, 1999; Cardy, 2008) or other interacting particle systems available in the physics literature (e.g. Escaff et al., 2018; Hernández-García and López, 2004).

### 5.3 Observation-scale dependence of the flux distribution

Phrasing the transport rate in terms of the Lagrangian dynamics of individual grains produces a flux distribution that adjusts with the observation time. According to Eq. (22), this adjustment is largely controlled by the deposition rate and Péclet number *Pe*. The deposition rate controls the duration of motions, while the Péclet number characterizes the significance of velocity fluctuations. Typical values for the Péclet number can be estimated from experimental publications as the ratio of mean and root-mean-square streamwise velocities of grains, giving $Pe = 0.8$ (Fathel et al., 2015), $Pe = 1.4$ (Heyman et al., 2016), $Pe = 3.7$ (Martin et al., 2012). For this range of Péclet numbers, Fig. 4 leads us to expect a monotonic decay of the mean flux with no obvious plateau at moderate observation times. Such a decay has been observed in several studies. Both Bunte and Abt (2005) and Singh et al. (2009) noted a power-law decay to a constant flux with observation time, in field and laboratory data respectively. However, these studies also observed that at high transport rates, the flux *increased* with observation time, rather than decreased. Possibly, at higher transport rates, collective rather than grain scale processes may principally control the scaling of the mean flux with observation time.

Both Singh et al. (2009) and Saletti et al. (2015) investigated higher statistical moments of the bedload flux beyond the mean in laboratory data. They found that higher moments also shift with the observation time $T$, but Singh et al. identified statistical *multiscaling*, where the flux distribution changes shape with $T$, while Saletti et al. identified *monoscaling*, where the distribution does not change shape with $T$. Our analysis predicts monoscaling because the flux distribution (18) is always Poissonian, even though all of its moments scale together with $T$ in a non-trivial way, see Fig. (4). Probably, the Poissonian and monoscaling characteristics of the flux distribution both originate from the assumed independence of individual particle motions, not from the distributions of velocities, movement times, or resting times. In particular, Eq. 18 indicates that changing the force $F(v)$ to produce exponential velocities will only

modify $\Lambda(T)$, not the shape or scaling type (mono, multi) of the flux distribution. Possibly, wider-tailed, multiscaling sediment flux distributions will derive from generalizations of our approach to include particle-particle interactions. Turowski (2010) demonstrated that renewal theories with certain non-exponential inter-arrival times produce multiscaling, and it is likely that non-exponential distributions will originate from particle-particle interactions (cf. Ancey et al., 2008). To some extent, a flux distribution which is wide-tailed at short observation times *must* be multiscaling, since it should approach the thin-tailed Einstein distribution (24) as the observation time becomes large, changing shape as it adjusts.

## 6 Conclusions

We have formulated the bedload flux probability distribution from the statistical mechanics of individual grains in transport. This formulation produces Poissonian flux distributions having scale-dependent rates, meaning transport rate fluctuations are relatively narrow, and transport characteristics shift with the timescales over which they are observed. In laboratory experiments, sediment transport fluctuations are typically wider than Poissonian. Notably, we can assert that the Poisson flux distribution derived in this paper originates exclusively from the independence of individual grains: the Poisson form is completely indifferent to the forces driving particles downstream, so long as these forces do not introduce correlations between particles. In the future, it will be necessary to refine the statistical mechanics formulation presented here to produce wider transport fluctuations. We expect that introducing any component in Eq. (5) which couples one particle to another will achieve wider flux distributions than Poissonian. The severe challenge will be evaluating the average in Eq. (13) when grains are not independent.

## Appendix A: Derivation of the phase space master equation

Because the joint process $\alpha = (x, v, \sigma)$ is Markovian, its phase space distribution function for a particular realization of the white noise $\xi(t)$ obeys the Chapman-Kolmogorov equation (Cox and Miller, 1965; Van Kampen, 2007):

$$W_\xi(\alpha, t + \delta t | \alpha_0) = \int d\alpha' W_\xi(\alpha, t + \delta t | \alpha', t) W_\xi(\alpha', t | \alpha_0).$$
(A1)

Here, $\int d\alpha = \sum_{\sigma=0,1} \int_{-\infty}^{\infty} dx \int_{-\infty}^{\infty} dv$. This equation relates the phase space distribution function at $t + \delta t$ to its value at $t$ through the transition amplitudes $W_\xi(x, v, \sigma, t + \delta t | x', v', \sigma', t)$. The distribution $W_\xi$ is a *functional* of the white noise $\xi(t)$ (Hänggi, 1985; Łuczka, 2005).

In the limit of vanishing $\delta t$, the transition amplitudes in Eq. (A1) can be directly evaluated from the dynamical equations Eq. (5) using a method analogous to Horsthemke and Lefever

(1984). The transition rates involve $\delta$-function terms in $x'$ and $v'$. These terms are expanded in $\delta t$ to first order, and the integrals in Eq. (A1) are conducted over the $\delta$-functions. This produces the pair of equations

$$\partial_t M_\xi = k_E R_\xi - k_D M_\xi + \big\{ -\partial_x v + \partial_v[-F(v) + \xi(t)]\big\} M_\xi$$
$$\partial_t R_\xi = k_D M_\xi - k_E R_\xi.$$
(A2)

In these equations the shorthands are $M_\xi = W_\xi(x,v,1,t|x_0,v_0,\sigma_0,t_0)$ and $R_\xi = W_\xi(x,v,0,t|x_0,v_0,\sigma_0,t_0)$. We now average Eq. (A2) over realizations of the white noise and compute the correlator $\langle \xi(t) M_\xi \rangle = \Gamma \partial_v M$ using the Furutsu-Novikov theorem (e.g. Van Kampen, 2007; Gitterman, 2005). Incorporating this correlator into Eq. (A2) obtains

$$\partial_t M = k_E R - k_D M + \hat{L}_K M$$
$$\partial_t R = k_D M - k_E R,$$
(A3)

where $\hat{L}_K = -\partial_x v + \partial_v\{-F(v) + \Gamma \partial_v\}$ is the Kramers operator, $M = \langle M_\xi \rangle$, and $R = \langle R_\xi \rangle$. Summing the above equations, noting $W(x,v,t|0) = M + R$, and eliminating $M$ from the sum produces Eq. (7).

### Appendix B: Solution for the position probability distribution

The position probability distribution can be obtained from Eq. (11) provided we have a pair of initial conditions on $P$. We consider that particles have a probability $k_D/k = \varphi$ to start from rest, so they have a probability $1 - \varphi = k_E/k$ to start from motion. Particles are initially located at $x = 0$, and particles that start from motion are considered to have a random initial velocity selected from the steady-state distribution

$$f(v) = \sqrt{\frac{\gamma}{2\pi\Gamma}} e^{-\gamma v^2/2\Gamma}.$$
(B1)

With these assumptions, the initial state can be written $M(x,v,0) = (1-\varphi)\delta(x)f(v)$ and $R(x,v,0) = \varphi\delta(x)\delta(v)$. Summing these two equations and integrating out the velocity provides $P(x,0) = \delta(x)$. Plugging these two equations into Eq. (A3), summing the result, then integrating out the velocity provides $\partial_t P(x,0) = -\frac{k_E V}{k}\delta'(x)$. This produces the required pair of initial conditions. A similar calculation is available in Weiss (2002).

Now we take Fourier transforms over space and Laplace transforms over time of the overdamped master equation (11), obtaining

$$\bar{\tilde{P}}(g,s) = \frac{s + k + Dg^2 - igV\varphi}{s(s+k) + (Dg^2 - igV)(s + k_E)}.$$
(B2)

The Fourier transform can be inverted by partial fraction decomposition and contour integration to obtain

$$\tilde{P}(x,s) = \frac{-\varphi D \partial_x^2 + V\varphi \partial_x + s + k}{VQ(s + k_E)} \exp\left[\frac{Vx}{2D} - \frac{V|x|}{2D}Q\right],$$
(B3)

where

$$Q = \sqrt{1 + \frac{4D}{V^2}\frac{s(s+k)}{s + k_E}}.$$
(B4)

The Laplace transform can then be inverted with the shift property $\mathcal{L}^{-1}\{\tilde{f}(s+k)\} = e^{-kt} f(t)$, the derivative property (Arfken, 1985)

$$\mathcal{L}^{-1}\{s\tilde{f}\} = (\delta(t) + \partial_t)f(t),$$
(B5)

the Bessel function identity (Bateman and Erdelyi, 1953, pg. 133)

$$\mathcal{L}^{-1}\left\{\frac{1}{s}\tilde{g}(s - a/s)\right\} = \int_0^t \mathcal{I}_0\left(2\sqrt{au(t-t')}\right)g(t')dt', \quad \text{(B6)}$$

and known Laplace transform pairs (Prudnikov et al., 1992), eventually giving Eq. (19).

### Appendix C: Calculation of the scale-dependent rate function

The Laplace transform of Eq. (17) over $T$ provides

$$\tilde{\Lambda}(s) = \rho \int_0^\infty dx_i \int_0^\infty dx \tilde{P}(x + x_i, s).$$
(C1)

Noting that $x + x_i$ is always positive, inserting Eq. (19), and integrating twice gives

$$\tilde{\Lambda}(s) = -\frac{\rho\varphi D}{VQ(s + k_E)} + \frac{2\rho D\varphi}{VQ(1-Q)(s + k_E)}$$
$$+ \frac{4\rho D^2(s+k)}{V^3 Q(1-Q)^2(s + k_E)}. \quad \text{(C2)}$$

Taking the inverse transform, applying Eq. (B5), and using the shift property develops

$$\Lambda(T) = \rho e^{-k_E T}\mathcal{L}^{-1}\left\{ -\frac{\varphi D}{VQ_\star s} + \frac{2D\varphi}{VQ_\star(1-Q_\star)s} \right.$$
$$\left. + \frac{4D^2(\bar{\partial}_T + k)}{V^3 Q_\star(1-Q_\star)^2 s} \right\}. \quad \text{(C3)}$$

Here, the notation $\bar{\partial}_T$ means the derivative acts from the left on all terms multiplying it, and

$$Q_* = \sqrt{1 + \frac{4D(k_D - k_E)}{V^2} + \frac{4D}{V^2}\left(s - \frac{k_E k_D}{s}\right)}.$$
(C4)

Laplace inverting Eq. (C3) with Eq. (B6) and tabulated Laplace transforms (e.g. Arfken, 1985; Prudnikov et al., 1992) eventually provides Eq. (21).

## Appendix D: Asymptotic limits of the flux rate function

The behavior of Eq. (21) at extreme values of $T$ can be obtained with Tauberian theorems by inverting the Laplace-transformed rate function (C2) at the opposite extreme of $s$ (Weiss, 1994). At short times, expanding Eq. (C2) as $s \to \infty$ gives

$$\tilde{\Lambda}(s) = \frac{\rho k_E V}{2k s^2} + \frac{\rho k_E}{k} \sqrt{\frac{D}{4s^3}}, \qquad (D1)$$

which inverts to

$$\Lambda(t) \sim \frac{\rho k_E V T}{2k} + \frac{\rho k_E}{k} \sqrt{\frac{DT}{\pi}}, \qquad (D2)$$

giving the small $T$ behavior. This has two scaling limits within it. Provided that $T \ll 2D/V^2$, the scaling goes as $\Lambda(T) \sim T^{-1/2}$. But if $T \gg 2D/V^2$, it goes as $\Lambda(t) \sim T$. For large times, taking $s \to 0$ gives

$$\tilde{\Lambda}(s) = \frac{\rho k_E V}{k s^2}, \qquad (D3)$$

and this inverts to $\Lambda(T) = \rho k_E V T/k$. These limits are summarized in Eq. (22).

**Code availability.** Python scripts used for the Monte Carlo simulation of Eq. (5) and to generate all figures have been made publically available at DOI: 10.5281/zenodo.6573311 . The scripts contain comments detailing the stochastic simulation methods.

**Author contributions.** All authors (KP, MH, and RF) contributed equally to ideation and manuscript preparation. KP performed all calculations and constructed all figures.

**Competing interests.** We declare no competing interests.

**Acknowledgements.** We would like to thank the two anonymous reviewers for their careful reviews that improved the manuscript.

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
