# Peer review of "A probabilistic description of bedload fluxes from the aggregate dynamics of individual grains"

_Earth Surface Dynamics, 2022_

## Author Comment (AC1)

**Response to Reviewer # 1**

Kevin Pierce, Marwan Hassan, Rui Ferreira

We would like to thank the first reviewer for their detailed comments on the manuscript. We have prepared revisions in response to all of the major points.

1. "From this perspective, the present formulation is intrincically the same as that usedis previously, for example, by Fan et al. (2014), who considered the motions mechanically, while simulated the transport process by switching the motions of the particle on and off (Fan et al., 2016). The authors may need to discuss this point explicitly."

   The underlying idea of particle movements as a Langevin equation while the rests are treated heuristically is certainly contained in *Fan et al.* [2016]. We will indicate this in the revised manuscript.

2. "Keep this in mind, the starting point of this work, Eq. (5), can only be considered as a "formal description", because the entrainment and deposition of the grain are not formulated mechanically. That is, the start and end of the motions of a particle are not determined by the forces acting on it; thus no new information on the travel and resting times can be obtained based on incorporating this dichotomous Markov noise."

   In the revised manuscript we will try to be more clear on the point that the motion-rest alternation is not treated mechanistically. A key contribution of the manuscript is to formalize the heuristic motion-rest alternation used in random-walk models like *Lajeunesse et al.* [2017] using the dichotomous Markov noise concept from the physics literature. We couple this formalism with descriptions of the movement phase based on Langevin models. This allows us to write probabilistic equations and find analytical solutions – additional benefits compared to *Fan et al.* [2016]. Perhaps in the future it will be possible to derive the dichotomous noise as an approximation of more detailed physics. We believe the manuscript establishes a starting point for this task.

3. "The authors are also suggested to discuss the effects of the velocity distributions on their deduced results."

   The Poissonian and monoscaling characteristics of the flux distribution both originate from the assumed independence of individual particle motions which was used to arrive at Eq. (16). They do not relate to the movement or resting time distributions or the particular forcing terms included in the Langevin model. We will attempt to better describe this in the revised manuscript.

4. "Wu et al. (2020) provided an explanation for the existence of the two different distributions, by pointing out that the long trajectories contribute to the Gaussian velocities, and the mixture of both long and short trajectories results in the exponential distribution; the long and short trajectories are distinguished by the shift of the hop distance-time scaling. Resorting to this result I think is important for clarifying some key issues in this work..."

   We agree it is worth describing that *Wu et al.* [2020, 2021] have attributed the shape of the velocity distribution to the balance of long and short hops in the velocity statistics. However this does not seem to be the last word on this problem. Another (probably related) view is that the shape of the velocity distributions relates to the amount of momentum dissipated by particle-bed collisions: this has been demonstrated to predict all distributions which have been reported so far from experiments, including the less common Gamma-like ones [*Pierce*, 2021]. Alternative explanations may still be possible, perhaps involving the interactions of particles with the vertical flow structure.

5. "For the "overdamped" approximation, explained by the authors as "moving particles attain their steady-state velocities relatively quickly after entrainment", which is only valid for the description of the long trajectories of particle motions. This is because only the long trajectories have a well defined mean velocity (e.g. the "steady-state velocity"); and the mean velocity for the short trajectories can on the average increase with their travel times (Wu et al., 2020). Since the short trajectories can cover over 80% of the total trajectories in experiments (Wu et al., 2021), applying this "overdamped" approximation may not be appropriate."

    We agree that the overdamped approximation should not be valid for exponential velocities. In fact one needs a forcing term that is linear in velocity to conduct an overdamped approximation whether it is approximately valid or not, see *Risken* [1984]. We plan to add an an explicit statement that the overdamped approximation is only possible for Gaussian velocities in the revised manuscript.

6. "There are recent studies using different methods to theretically address the motion period of the bedload particle transport, for example, as discussed above (Wu et al., 2020; Wu et al., 2021), the results of which are compared with measured data. In other words, how the particle velocity changes with time was proposed and further determined based on experimental measurements (i.e. other means of specifying the external forces acting on the particle, F(u) in this work). The authors can compare the part of their formulation on the particle motions with different results."

    Given that our formulation spans the local, intermediate, and global ranges of *Nikora et al.* [2001], while no one dataset also spans these ranges [e.g. *Pierce and Hassan*, 2020; *Pretzlav et al.*, 2021], a comparison to experimental data would be a challenging data compilation problem. All components of this work have however been tested independently in other studies: *Ancey and Heyman* [2014] and *Heyman et al.* [2016] tested the velocity model (2) for moving particles, while *Lajeunesse et al.* [2017] tested the motion-rest alternation component (1) at the global range. For this reason we have chosen to present a purely formal description that combines and generalizes earlier stat mech descriptions of bedload transport.

7. "Could the derivation be started directly from the probability distribution function based on the continuum master equation (5)?"

    We could certainly use the joint distribution functions as the starting point for the calculation of the flux, rather than Eq. 13, but we prefer to emphasize the particle-scale origins of the flux, wherein particle concentrations are represented as arrays of discrete points using indicator functions, and probability distributions result from ensemble averages over these indicators.

**References**

Ancey, C., and J. Heyman, A microstructural approach to bed load transport: Mean behaviour and fluctuations of particle transport rates, *Journal of Fluid Mechanics*, *744*(2014), 129–168, doi:10.1017/jfm.2014.74, 2014.

Fan, N., A. Singh, M. Guala, E. Foufoula-Georgiou, and B. Wu, Exploring a semimechanistic episodic Langevin model for bed load transport: Emergence of normal and anomalous advection and diffusion regimes, *Water Resources Research*, *52*, 2789–2801, doi:10.1002/2016WR018704.Received, 2016.

Heyman, J., P. Bohorquez, and C. Ancey, Entrainment, motion, and deposition of coarse particles transported by water over a sloping mobile bed, *Journal of Geophysical Research: Earth Surface*, *121*(10), 1931–1952, doi:10.1002/2015JF003672, 2016.

Lajeunesse, E., O. Devauchelle, and F. James, Advection and dispersion of bed load tracers, *Earth Surface Dynamics*, *6*(2), 389–399, doi:10.5194/esurf-6-389-2018, 2017.

Menzel, A. M., and N. Goldenfeld, Effect of Coulombic friction on spatial displacement statistics, *Physical Review E*, *84*(1), 1–9, doi:10.1103/PhysRevE.84.011122, 2011.

Nikora, V., J. Heald, D. Goring, and I. McEwan, Diffusion of saltating particles in unidirectional water flow over a rough granular bed, *Journal of Physics A: Mathematical and General*, *34*(50), doi:10.1088/0305-4470/34/50/103, 2001.

Pierce, J. K., The stochastic movements of individual streamed grains, Ph.D. thesis, The University of British Columbia, 2021.

Pierce, J. K., and M. A. Hassan, Back to Einstein: Burial-induced three-range diffusion in fluvial sediment transport, *Geophysical Research Letters*, *47*(15), doi:10.1029/2020GL087440, 2020.

Pretzlav, K. L., J. P. Johnson, and D. N. Bradley, Smartrock transport from seconds to seasons: Shear stress controls on gravel diffusion inferred from hop and rest scaling, *Geophysical Research Letters*, *48*(9), 1–11, doi:10.1029/2020GL091991, 2021.

Risken, H., *The Fokker-Planck Equation: Methods of Solution and Applications*, 2nd ed., Springer-Verlag, Ulm, doi:10.1080/713821438, 1984.

Wu, Z., D. Furbish, and E. Foufoula-Georgiou, Generalization of hop distance-time scaling and particle velocity distributions via a two-regime formalism of bedload particle motions, *Water Resources Research*, *56*(1), 1–30, doi:10.1029/2019WR025116, 2020.

Wu, Z., A. Singh, E. Foufoula-Georgiou, M. Guala, X. Fu, and G. Wang, A velocity-variation-based formulation for bedload particle hops in rivers, *Journal of Fluid Mechanics*, *912*, doi:10.1017/jfm.2020.1126, 2021.

---

## Author Comment (AC2)

**Response to Reviewer # 2**

Kevin Pierce, Marwan Hassan, Rui Ferreira

Thank you for the detailed and constructive comments. We plan to revise the manuscript to address every point as we discuss below.

**1   Major comments**

1. " The paper is well written, clear and concise. The approach is sound and standard mathematical tools are briefly introduced before or after they are used, which helps to understand the main ideas behind technical derivations. Main equations however miss a detailed physical explanation, term by term, to be understood by the readership."

   Thanks - we took for granted that the basic structure of mobile-immobile models and the role of advection and diffusion terms is understood by the reader. To make the manuscript more approachable we will add more detailed term-by-term descriptions for major equations in the revised manuscript.

2. "The title should be more precise, several stochastic description of bedload having been already proposed."

   This is helpful advice. As you pointed out, one of the key contributions in the manuscript is the simultaneous treatment of motion-rest exchange and velocity fluctuations, so we will emphasize this in a revised title.

3. "A general concern is that the stochastic approach, although theoretically sound, is weakly linked to actual statistics of sediment transport by bedload, and thus the relevance of such complicated form of the bedload flux (eq 21!)  is questionable for realistic transport conditions. In particular, there is no discussion on the actual values of Péclet number and the importance of considering both velocity fluctuations and entrainment/deposition as processes acting on similar time scales. There are considerable simplifications when decoupling both, so the authors should better point why such coupled approach is necessary. By doing so, the authors should also consider comparing their results with existing experimental or numerical data."

   We can estimate typical values of the Péclet number from the available experimental data and include these in the revised manuscript. Justifying the decoupling between motion-rest alternation and particle movement velocities is more difficult with existing data. One of us (Ferreira) has unpublished experimental data showing that the acceleration phase following entrainment is commonly more than ten times shorter than the period particles spend in motion. A similar conclusion is described in *Campagnol et al.* [2015]: here the acceleration phases following entrainment and preceding deposition are reported as commonly shorter than the durations of intermediate trajectories. We will include this citation in the revised manuscript to support our approximation. In any case our manuscript presents the first analytical model which describes particle entrainment and deposition simultaneously with particle motion – even if they are decoupled. For this reason we believe the manuscript will be important for any future modeling studies seeking to couple motion-rest exchange to the particle motion.

**2   Minor comments**

1. 12 : drop "really", and precise why /when fluctuations matter ?

Thanks, this is good advice. We will add content on the role of fluctuations in bedform initiation, the difficulty they present for transport prediction, and their emphasized magnitude in weak transport conditions.

2. 16 : What is a "classic" description ? Deterministic ?

   By "classic" we meant "of recognized and established value". We will change this to "deterministic" as it conveys the same point and avoids confusion.

3. 17-19 : I do not get the point here. The approach followed by the authors is also mainly kinematic in that no discussion is made on the forces (gravitational, drag, friction, collision,...) acting on particles.

   We are using modeled forces on particles to predict dynamics from Newtonian equations. This is distinctly different from CTRW approaches, for example, where modeled forces do not enter the description. The advantage is that we can change the modeled forces to represent friction, drag, etc. without changing the essential structure of the description. In particular, Eq. 7 follows for whatever choice of $F(v, t)$ in Eq. 5. For example, the results of *Lajeunesse et al.* [2017] follow for the choice $F(v) = $ constant. Later, we plan to explore episodic collision forces as in *Pierce* [2021].

4. The original "probabilistic" description ...

   We are happy to add "probabilistic". We are however not aware of an earlier model of grain-scale displacements than *Einstein* [1937].

5. 21 Later → replace by "more recently" (there were a lot a probabilistic studies between Einstein and Lisle)

   This is a fair point! We will modify.

6. 22 "by promoting his instantaneous steps to intervals of motion with constant velocity" I do not get the meaning of promoting hear.

   "Replacing" works just as well.

7. 75 - 85 No mention of Continuous Time Random Walks model is made. Authors should compare their approach with for instance [Schumer et al 2009]

   Our emphasis is on a more detailed treatment of the movement phase in a mobile-immobile model, so our stepping off point is from the two-state CTRWs like *Lajeunesse et al.* [2017], not the one-state CTRWs described in [*Schumer et al.*, 2009]. There are careful discussions on linkages between these models in our earlier works [*Pierce and Hassan*, 2020; *Pierce*, 2021]. We omitted this discussion here for brevity, but we can add the key CTRW references in the revision.

8. l115 : Better explain how this equation can be physically understood, notably the presence of k and ke with time derivatives.

   The structure $\partial_t(\partial_t + k)W = (\partial_t + k_E)W$ encodes an intermittency factor at long times. To see this, take small $t$: one can then neglect the low order time derivatives and integrate over time ($\int_{-\infty}^{t} dt$) to obtain $\partial_t W = LW$. From this we conclude that at short times, particles obey the Kramers equation. Now take large $t$. One now neglects the high order time derivatives to get $\partial_t W = \frac{k_E}{k} LW$. This is a Kramers equation with its velocity and position evolution *slowed* by the intermittency factor $k_E/k$ – the fraction of time the particle spends in motion. We will adapt this explanation to the revised manuscript.

9. l137 Is the overdamped approximation similar to adiabatic elimination of the fast variable ? A deeper discussion is needed here, notably the validity of such approximation with respect to typical bedload transport scales.

Yes, the analogy is to take the velocity as the fast variable. There are some papers by Van Kampen that discuss this point. We will add a more detailed discussion of the validity of the overdamped approximation using the discussion on bedload transport timescales in *Campagnol et al.* [2015].

10. l 143 : How does such expression compare with a spatio-temporal markov process, for instance eq 4.4 in Ancey & Heyman JFM 2014 ?

    The short answer is that they are not directly comparable because $P(x, t)$ is a single-particle density and not the particle activity which is considered in Ancey and Heyman. This said, there may be a way to relate these formulations by coarse graining our formulation. This will be something to investigate in the future.

11. l217 : Why would velocity fluctuation during motion decrease diffusion at small time scales ? I would have imagined the reverse.

    Thanks, this is a typo. As one can see in the Fig. 3b, velocity fluctuations *increase* the diffusion at short timescales compared to the analytical formula that neglects velocity fluctuations.

12. l230 Rewriting the Péclet in its usual form (diffusion time scale over advection time scale) would help understanding the transport process the authors are trying to characterize. In there definition of Peclet, the important length scale is the mean particle jump length. This should appear somewhere.

    Here the timescales are $\tau_a = V/k_D$ for advection and $\tau_d = \sqrt{2D/k_D}$ for diffusion – the displacements due to each process in the mean movement time $1/k_D$. Our Péclet number is then $Pe = (\tau_a/\tau_d)^2$. We'll add a mention of these timescales.

13. l264 : can you give an physical interpretation of why the flux is higher at the beginning ? Do we have a higher probability to sample particles in motion at short time scales ?

    The probability to sample particles in motion is constant through time given our chosen initial conditions in Appendix B. The flux is dominated by particle velocity fluctuations at short timescales, i.e. particles that transport much faster than the mean, whereas it is dominated by the ensemble mean velocity at long timescales, i.e. when the influence of velocity fluctuations from individual particles averages out. For this reason, the peak in the flux as $t \to 0$ grows as the Péclet number shrinks. We will add a statement to this effect in the revised manuscript.

14. Figure 4a : why is there a plateau between 1-100 s? Is the mean flux only dependent on Péclet and observation time ? If yes, can you make it appear clearly in eq 21. If not, what are the fixed parameter in this figure ? Can you compare with experimental/numerical (DEM) data ?

    We can infer from the manuscript equations that the *asymptotic* values of $\langle q(T)\rangle/q_0$ only depend on $Pe$ and $k_D T$, but this says nothing about the intermediate ("plateau") region, which may depend on the other parameters. We have

$$\frac{q}{q_0} \sim \begin{cases} \sqrt{\frac{Pe}{2\pi k_D T}}, & T \ll (k_D Pe)^{-1} \\ 1, & T \gg (k_D Pe)^{-1} \end{cases}. \tag{1}$$

    We chose not to plot figure 4a using the dimensionless $k_D T$ because all curves then partially overlap, which obscures the patterns (like the emergence of a plateau) and makes for challenging visual comparison between the analytical and numerical results. We will clearly state which parameters are modified and which are held constant in a revised figure caption.

    At intermediate times, we suspect the plateau emerges (when the diffusion is relatively strong) because velocity fluctuations continue to control the flux, but both negative and positive excursions of velocity become relevant – Particles that are both slower and faster than the mean can cross the control surface. However we are not clear on this point, so we chose not to include it in the paper.

15. Figure 4b If the distribution is Poissonian, you should be able to rescale it by its mean and have a single time-independent distribution. Could you plot this ?

Yes, we can certainly rescale, but we decided not to plot the distribution in this way in favor of demonstrating how the distribution width changes with observation time in Fig. 4b. We preferred to demonstrate in Fig. 4b how the *Einstein* [1950] theory gradually emerges as $T \to \infty$. Probably we can add a sentence indicating that the distribution collapses with the mean.

**References**

Campagnol, J., A. Radice, F. Ballio, and V. Nikora, Particle motion and diffusion at weak bed load: Accounting for unsteadiness effects of entrainment and disentrainment, *Journal of Hydraulic Research*, *53*(5), 633–648, doi:10.1080/00221686.2015.1085920, 2015.

Einstein, H. A., Bed load transport as a probability problem, Ph.D. thesis, ETH Zurich, 1937.

Einstein, H. A., The bedload function for sediment transportation in open channel flows, *Tech. rep.*, United States Department of Agriculture, Washington, DC, 1950.

Lajeunesse, E., O. Devauchelle, and F. James, Advection and dispersion of bed load tracers, *Earth Surface Dynamics*, *6*(2), 389–399, doi:10.5194/esurf-6-389-2018, 2017.

Pierce, J. K., The stochastic movements of individual streambed grains, Ph.D. thesis, The University of British Columbia, 2021.

Pierce, J. K., and M. A. Hassan, Back to Einstein: Burial-induced three-range diffusion in fluvial sediment transport, *Geophysical Research Letters*, *47*(15), doi:10.1029/2020GL087440, 2020.

Schumer, R., M. M. Meerschaert, and B. Baeumer, Fractional advection-dispersion equations for modeling transport at the Earth surface, *Journal of Geophysical Research: Earth Surface*, *114*(4), 1–15, doi: 10.1029/2008JF001246, 2009.

---

## Author Response (AR1)

**Response to Reviewers: ESurf submission "Stochastic description of intermittent transport and aggregate derivation of the bedload flux" Kevin Pierce, Marwan Hassan, Rui Ferreira**

We would like to thank both of the anonymous reviewers for their detailed comments on the manuscript. We have made comprehensive changes in response to every reviewer comment. Below we have explained our revisions. We hope the revised manuscript is satisfactory to the Editors.

**Response to Reviewer #1**

1. "From this perspective, the present formulation is intrinsically the same as that used previously, for example, by Fan et al. (2014), who considered the motions mechanically, while simulated the transport process by switching the motions of the particle on and off (Fan et al., 2016). The authors may need to discuss this point explicitly."

Please see the revised lines 143-149, where we explain precisely the relationship of our work to *Fan* et al. [2014, 2016].

2. "Keep this in mind, the starting point of this work, Eq. (5), can only be considered as a "formal description", because the entrainment and deposition of the grain are not formulated mechanically. That is, the start and end of the motions of a particle are not determined by the forces acting on it; thus no new information on the travel and resting times can be obtained based on incorporating this dichotomous Markov noise."

We agree completely and have added an explicit statement mirroring this perspective at L357-359.

3. "The authors are also suggested to discuss the effects of the velocity distributions on their deduced results."

We have added explicit discussion on the (lack of) effects of the velocity distributions at lines 408-409

4. "Wu et al. (2020) provided an explanation for the existence of the two different distributions, by pointing out that the long trajectories contribute to the Gaussian velocities, and the mixture of both long and short trajectories results in the exponential distribution; the long and short trajectories are distinguished by the shift of the hop distance-time scaling. Resorting to this result I think is important for clarifying some key issues in this work..."

We have discussed the analyses of Wu et al. [2020] and Wu et al. [2021] at lines 49-51 alongside the complementary perspective of *Pierce* [2021].

5. "For the "overdamped" approximation, explained by the authors as "moving particles attain their steady-state velocities relatively quickly after entrainment", which is only valid for the description of the long trajectories of particle motions. This is because only the long trajectories have a well defined mean velocity (e.g. the "steady-state velocity"); and the mean velocity for the short trajectories can on the average increase with their travel times (Wu et al., 2020). Since the short trajectories can cover over 80% of the total trajectories in experiments (Wu et al., 2021), applying this "overdamped" approximation may not be appropriate."

We have added explicit statements that the overdamped approximation is only possible for Gaussian velocities at lines 187-189 and 325-326. Additionally, we have added additional citations and discussion to support the overdamped approximation and explain the conditions when the acceleration phase might be acceptably neglected at lines 190-194 and 336-337.

Additionally, we have unpublished data showing that the overdamped approximation is reasonable for 5mm glass beads in transport. We have attached a figure summarizing these data in the response to reviewer #2, which can be found below.

6. "There are recent studies using different methods to theoretically address the motion period of the bedload particle transport, for example, as discussed above (Wu et al., 2020; Wu et al., 2021), the results of which are compared with measured data. In other words, how the particle velocity changes with time was proposed and further determined based on experimental measurements (i.e. other means of specifying the external forces acting on the particle, F(u) in this work). The authors can compare the part of their formulation on the particle motions with different results."

We added citations to *Campagnol et al.* [2015] in several locations– lines 190 and 305: this paper is the only one to our knowledge that examines the evolution of the velocity statistics through time from entrainment.

7. "Could the derivation be started directly from the probability distribution function based on the continuum master equation (5)?"

We could certainly use the joint distribution functions as the starting point for the calculation of the flux, rather than Eq. 13, but we preferred to emphasize the particle-scale origins of the flux, wherein particle concentrations are represented as arrays of discrete points using indicator functions, and probability distributions result from ensemble averages over these indicators.

Thank you sincerely for your effort reading and commenting on the manuscript. We hope that we have incorporated your suggestions to your satisfaction.

**Response to Reviewer # 2**

**Major comments**

1. "The paper is well written, clear and concise. The approach is sound and standard mathematical tools are briefly introduced before or after they are used, which helps to understand the main ideas behind technical derivations. Main equations however miss a detailed physical explanation, term by term, to be understood by the readership."

Please see lines 205-209 and 161-172, where we have added detailed descriptions of the individual terms in the equations.

2. "The title should be more precise, several stochastic description of bedload having been already proposed."

Thanks - we changed the title to "Stochastic description of intermittent transport and aggregate derivation of the bedload flux". We hope this is sufficiently descriptive. Thanks for your advice on this.

3. "A general concern is that the stochastic approach, although theoretically sound, is weakly linked to actual statistics of sediment transport by bedload, and thus the relevance of such complicated form of the bedload flux (eq 21!) is questionable for realistic transport conditions. In particular, there is no discussion on the actual values of Péclet number and the importance of considering both velocity fluctuations and entrainment/deposition as processes acting on similar time scales. There are considerable simplifications when decoupling both, so the authors should better point why such

Figure 1: This figure demonstrates that the acceleration phase of particles following entrainment is typically short compared to the duration of trajectories.

coupled approach is necessary. By doing so, the authors should also consider comparing their results with existing experimental or numerical data."

We have provided estimates of the Péclet numbers seen in experiments in a new discussion paragraph at L390. We added discussion on the validity of neglecting the acceleration/deceleration phases of particle motions using the results of *Campagnol et al.* [2015] at L190-195.

We mentioned that we have some unpublished experimental data showing that the acceleration phase following entrainment is commonly more than ten times shorter than the period particles spend in motion. Please see the below figure. This figure shows the trajectories of 5mm glass beads in relatively weak transport which show Gaussian velocity distributions. The acceleration phase is visible in the trajectories, and is relatively short compared to the full trajectory between entrainment and either deposition or departure from the viewing window. We plan to work on this project in more detail in the future.

**Minor comments**

1. 12 : drop "really", and precise why /when fluctuations matter ?

We emphasized that transport fluctuations are strongest in weak transport conditions that are characteristic of gravel-bed rivers in our edits at lines 14-19.

2. 16 : What is a "classic" description ? Deterministic ?

We replaced "classic" with "deterministic" at L22.

3. 17-19 : I do not get the point here. The approach followed by the authors is also mainly kinematic in that no discussion is made on the forces (gravitational, drag, friction, collision,...) acting on particles.

We have added additional discussion of what exactly incorporating forces into the motion state gives us at L135-149. This is an important next step toward a fully mechanistic description of particle transport at the grain scale.

4. The original "probabilistic" description ...

We added "stochastic" at L30.

5. 21 Later  $\rightarrow$  replace by "more recently" (there were a lot a probabilistic studies between Einstein and Lisle)

We rephrased the entire paragraph, which now appears near L35, to indicate the entire progression of the research, rather than skipping over the intermediate advancements as we did in the original manuscript."

6. 22 "by promoting his instantaneous steps to intervals of motion with constant velocity" I do not get the meaning of promoting hear.

We replaced "promoting" with "replaces".

7. 75 - 85 No mention of Continuous Time Random Walks model is made. Authors should compare their approach with for instance [Schumer et al 2009]

We added discussion of CTRW approaches at lines 32-36.

8. 1115 : Better explain how this equation can be physically understood, notably the presence of k and ke with time derivatives.

We added a description of how the mixed order time derivatives encode intermittency at lines 164-172.

9. 1137 Is the overdamped approximation similar to adiabatic elimination of the fast variable? A deeper discussion is needed here, notably the validity of such approximation with respect to typical bedload transport scales.

We added additional discussion of the overdamped approximation with reference to *Campagnol et al.* [2015] and earlier Langevin models of bedload velocities at lines 190-194. See also the above figure.

10. l 143 : How does such expression compare with a spatio-temporal markov process, for instance eq 4.4 in Ancey & Heyman JFM 2014 ?

We have left this for future investigation. The challenge, as we described before, is that P(x,t) is a single-particle density and not the particle activity as considered in Ancey and Heyman [2014]. We are investigating the relationships between these formulations along the lines of Ballio et al. [2014].

11. l217 : Why would velocity fluctuation during motion decrease diffusion at small time scales ? I would have imagined the reverse.

We fixed this typo at lines 277-279.

12. l230 Rewriting the Péclet in its usual form (diffusion time scale over advection time scale) would help understanding the transport process the authors are trying to characterize. In there definition of Peclet, the important length scale is the mean particle jump length. This should appear somewhere.

We made a typo in our earlier reply to your comment. We consider *length-scales* of advection and diffusion. It is explained how the Péclet number emerges as a ratio of these lengthscales in the revised lines 290-295.

13. l264 : can you give an physical interpretation of why the flux is higher at the beginning ? Do we have a higher probability to sample particles in motion at short time scales ?

We added a physical interpretation of why the flux is higher at lines 334-337. We added additional qualitative comparison to experimental data in a new discussion paragraph at L390.

14. Figure 4a : why is there a plateau between 1-100 s? Is the mean flux only dependent on Péclet and observation time ? If yes, can you make it appear clearly in eq 21. If not, what are the fixed parameter in this figure ? Can you compare with experimental/numerical (DEM) data ?

We added some speculation on why a plateau emerges at lines 335-337, and we added a statement of which variables are held constant at the end of the Fig4 caption.

15. Figure 4b If the distribution is Poissonian, you should be able to rescale it by its mean and have a single time-independent distribution. Could you plot this ?

We added a sentence at L45 indicating that  $\Lambda(T)$  is the only parameter of the flux.

Thank you again! We are most thankful for your supportive comments on the manuscript.

**References**

- Ancey, C., and J. Heyman, A microstructural approach to bed load transport: Mean behaviour and fluctuations of particle transport rates, *Journal of Fluid Mechanics*, 744 (2014), 129–168, doi:10.1017/jfm.2014.74, 2014.
- Ballio, F., V. Nikora, and S. E. Coleman, On the definition of solid discharge in hydro-environment research and applications, *Journal of Hydraulic Research*, 52(2), 173–184, doi:10.1080/00221686.2013.869267, 2014.
- Campagnol, J., A. Radice, F. Ballio, and V. Nikora, Particle motion and diffusion at weak bed load: Accounting for unsteadiness effects of entrainment and disentrainment, *Journal of Hydraulic Research*, 53(5), 633–648, doi:10.1080/00221686.2015.1085920, 2015.
- Fan, N., D. Zhong, B. Wu, E. Foufoula-Georgiou, and M. Guala, A mechanistic-stochastic formulation of bed load particle motions: From individual particle forces to the Fokker-Planck equation under low transport rates, *Journal of Geophysical Research: Earth Surface*, 119(3), 464–482, doi:10.1002/2013JF002823, 2014.
- Fan, N., A. Singh, M. Guala, E. Foufoula-Georgiou, and B. Wu, Exploring a semimechanistic episodic Langevin model for bed load transport: Emergence of normal and anomalous advection and diffusion regimes, *Water Resources Research*, 52, 2789–2801, doi:10.1002/2016WR018704.Received, 2016.
- Pierce, J. K., The stochastic movements of individual streambed grains, Ph.D. thesis, The University of British Columbia, 2021.
- Wu, Z., D. Furbish, and E. Foufoula-Georgiou, Generalization of hop distance-time scaling and particle velocity distributions via a two-regime formalism of bedload particle motions, *Water Resources Research*, 56(1), 1–30, doi:10.1029/2019WR025116, 2020.
- Wu, Z., A. Singh, E. Foufoula-Georgiou, M. Guala, X. Fu, and G. Wang, A velocity-variation-based formulation for bedload particle hops in rivers, *Journal of Fluid Mechanics*, 912, doi:10.1017/jfm.2020. 1126, 2021.